# PIAST: Rapid Prompting with In-context Augmentation for Scarce Training data

## Abstract

LLMs are highly sensitive to prompt design, but handcrafting effective prompts is difficult and often requires intricate crafting of few-shot examples. We propose a fast automatic prompt construction algorithm that augments human instructions by generating a small set of few shot examples. Our method iteratively replaces/drops/keeps few-shot examples using Monte Carlo Shapley estimation of example utility. For faster execution, we use aggressive subsampling and a replay buffer for faster evaluations. Our method can be run using different compute time budgets. On a limited budget, we outperform existing automatic prompting methods on text simplification and GSM8K and obtain second best results on classification and summarization. With an extended, but still modest compute budget we set a new state of the art among automatic prompting methods on classification, simplification and GSM8K. Our results show that carefully constructed examples, rather than exhaustive instruction search, are the dominant lever for fast and data efficient prompt engineering. We will make code and data publicly available upon acceptance.

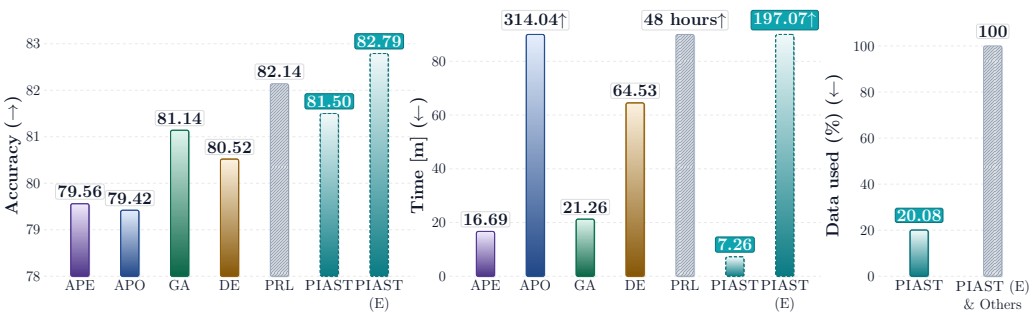

**Figure 1:** Overview of the results averaged over seven different text classification tasks, each run three times, comparing PIAST against current benchmarks. PIAST is able to generate high-quality prompts very efficiently, while requiring only a small portion of the dataset yielding comparable results to the current SOTA methods.

## 1 Introduction

Automatic prompt engineering has emerged as a practical way to adapt LLMs without gradient updates. However, many existing methods are impractical in time and data constrained settings: (i) some require hours of compute to explore a large prompt search space, and (ii) they rely on sizeable training sets to reliably score candidates. Moreover, most prior work optimizes only the instruction string, for example via rephrasing (using e.g. evolutionary algorithms or extensive search), ignoring the most impactful component of in-context learning (ICL): *few-shot examples*. The main exceptions are Batorski et al. (2025), which synthesizes examples, but its computational cost is dozens of hours, and Pryzant et al. (2023), which only selects examples from the training set.

Our method is, to our knowledge, the first method that is fast, synthesizes new few-shot examples not found in the training set and requires relatively less access to training examples. When run long enough, our method additionally obtains new state of the art results among automatic prompting methods for a number of tasks.

Our method works as follows: We first synthesize a proposal set of in-context examples that we append to our initial prompt. Then, in our optimization loop, we evaluate its efficacy on a small randomized evaluation set and identify the least helpful examples using a Monte Carlo Shapley estimator and replace, drop or keep it. If replaced, we draw from a pool of newly proposed few-shot examples. For efficiency and stability, we use a replay buffer for the evaluation set.

Our algorithm has a favorable anytime performance: When run with a small computational budget, we attain second best results among our baselines on classification and summarization and already exceeds previous SoTA on simplification and GSM8K. When run with an extended budget that is still comparable to some other baselines, we exceed previous methods additionally on classification. Interestingly, even when running without the iterative update loop and only using the first generated few-shot examples, we often still get competitive results.

To summarize, our contributions are as follows:

**Conceptual:** We propose PIAST, an automatic prompt construction method that augments a concise human-written instruction with a small set of automatically generated few-shot examples. We use an iterative improvement loop that improves the current set of few-shot examples using Shapley values to estimate utility of individual examples.

**Implementation:** For a fast implementation, we approximate Shapley values, KV-cache reuse for shared ICL prefixes, PagedAttention for compact KV memory management and continuous token-level batching to maintain high GPU utilization.

**Empirical Results:** We demonstrate strong performances using the same set of robust hyperparameters on text classification, summarization and simplification as well as GSM8K. Our approach yields strong anytime performance: When using only a subset of data and a small computational budget we obtain SoTA on text simplification and GSM8K and obtain second best results on summarization and classification. With an extended budget and full training set access we additionally set a new SoTA on classification among automatic prompting methods.

**Table 1:** Comparison of PIAST with other methods from the literature. Dataset indicates the fraction of the dataset used during construction of the prompt. Refinement shows whether the method iteratively improves the current prompt or generates a new one in a single step. Few-shot specifies whether the method is capable of generating few-shot examples. Auto Gen. denotes whether prompts are generated automatically.

| Algorithm | Dataset | Refinement | Few-shot | Auto Gen. | Speed |
|---|---|---|---|---|---|
| Manual Instruction Zhang et al. (2022) | ✗ | ✗ | ✗ | ✗ | ▮▮▮▮▮ |
| APE Zhou et al. (2022) | ✓ | ✗ | ✗ | ✓ | ▮▮▮▯▯ |
| APO Pryzant et al. (2023) | ✓ | ✓ | partial | ✓ | ▮▮▯▯▯ |
| EvoPrompt Guo et al. (2023) | ✓ | ✓ | ✗ | ✓ | ▮▮▮▯▯ |
| PRL Batorski et al. (2025) | ✓ | ✓ | ✓ | ✓ | ▮▯▯▯▯ |
| PIAST | partial | ✓ | ✓ | ✓ | ▮▮▮▮▯ |

**Legend:** ✓ yes ✗ no ⌇ partial  Speed: ▮▯▯▯▯ slow, . . . , ▮▮▮▮▮ fast.

## 2 RELATED WORK

**Prompt Engineering** improves model capabilities without retraining, keeping costs low (Liu et al., 2023). Chain-of-Thought (CoT) (Wei et al., 2022) elicits reasoning via intermediate steps; Tree-of-Thought (ToT) (Yao et al., 2023) explores multiple candidate paths, while Program-of-Thoughts (Chen et al., 2022) and Graph-of-Thoughts (Besta et al., 2024) structure prompts as pro-

grams and graphs. Least-to-Most prompting decomposes problems into simpler subproblems to strengthen compositional reasoning (Zhou et al., 2023); related advances include zero-shot CoT and self-consistency for more robust reasoning (Kojima et al., 2022; Wang et al., 2022). Few-shot prompting (Brown et al., 2020) conditions on in-prompt exemplars and is effective for puzzles and evidence extraction (Xu et al., 2023; Greenblatt, 2024; Sivarajkumar et al., 2024).

**Automated Prompt Engineering** aims to improve task performance by replacing manual prompt design with automated methods. The Automatic Prompt Engineer (APE) Zhou et al. (2022) generates candidate prompts from input–output examples and filters them based on performance. Since no benefits were observed from in-sample refinement, APE remains a purely generative approach. Pryzant et al. (2023) introduced Automatic Prompt Optimization (APO), which iteratively refines prompts using natural language critiques, effectively simulating a form of gradient descent. APO includes few-shot examples within its prompts, but it is limited to examples drawn from the training dataset. To improve efficiency, APO employs minibatching, beam search, and bandit selection. Guo et al. (2023) proposed EvoPrompt, which evolves a population of prompts using LLMs together with evolutionary operators, achieving strong results without requiring model gradients. Other approaches leverage reinforcement learning, such as RLPrompt Deng et al. (2022), which generates prompts of up to 5 tokens, and PRL Batorski et al. (2025), which synthesizes in-context examples autonomously whenever beneficial.

Among all these methods, only two are capable of incorporating examples into the prompt. The first is APO, which can only reuse examples from the training set, an inherent limitation on its performance. The second is PRL, which can introduce previously unseen examples, but at the cost of tens of hours of computation, making it impractical for scenarios where prompts must be produced quickly.

In contrast, our method PIAST can rapidly generate few-shot examples that are not present in the training data. We argue that the flexibility of deciding whether and which examples to include is directly tied to improved task performance. We give an overview of how PIAST compares to existing methods in Table 1.

## 3 METHOD

In this section, we present our method, which is composed of three components: the Prompt Proposer, the Prompt Evaluator, and the Prompt Improver. Each component is instantiated as a frozen LLM with a distinct role in the overall pipeline. Our final prompt consists of the hand-crafted instruction proposed by Zhang et al. (2022), concatenated with the in-context examples produced by our optimization procedure.

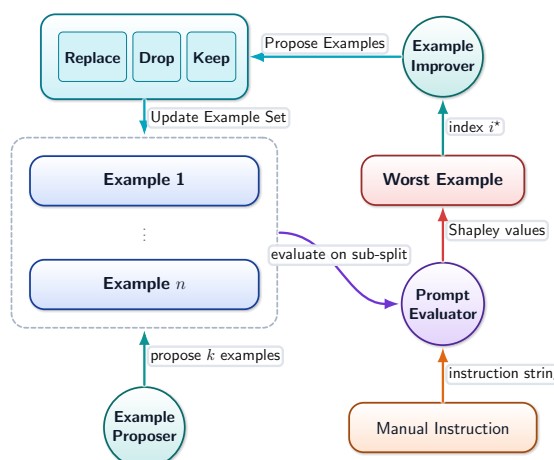

**Figure 2:** Pipeline of PIAST. Initially, the Example Proposer generates examples, which are then iteratively improved by evaluating them with the Prompt Evaluator and choosing new examples from the Example Improver to incorporate into the set of current in-context examples.

**Example Proposer.** The Example Proposer is responsible for generating initial candidate examples. It receives a task-specific initial instruction and produces a set of examples accordingly. To ensure coverage and robustness, the generated examples are deliberately diverse in both topic and length. Each example is then subject to a subsequent replace/drop/keep decision. The prompt for the Example Proposer is given in Appendix D.

**Prompt Evaluator.** The Prompt Evaluator assesses the quality of candidate prompts. Given a prompt, it evaluates its performance on a subset $D$ of the data using a specified metric $f$. This step ensures that only the most effective prompts are selected for further use. In our setup, the evaluated prompt is the base instruction concatenated with the proposed examples.

**Example Improver.** The Example Improver starts from a current prompt and iteratively changes the in-context examples in a replace/drop/keep cycle. The examples produced may differ in structure, topic, and length, thereby increasing diversity in the candidate pool. This process mirrors the behavior of the Prompt Proposer, where randomness in topic and sentence length is also introduced to encourage exploration. The prompt for the Example Improver is given in Appendix E.

The example improver uses Shapley values to estimate quality of examples and implements a replace/drop/keep cycle as detailed below.

**Example Selection via Shapley Values** Let $[n] = \{1, \ldots, n\}$ index the current few-shot examples and let $D$ denote the subset of training examples on which we evaluate. For any ordered subset $S = \{i_1 \prec \ldots \prec i_n\}$, where $i_1, \ldots, i_n \in [n]$, let $P(S)$ be the ICL prompt formed from concatenating the examples in $S$ according to their order. We define the utility of $S$ as the evaluator's accuracy on $D$:

$$v(S) \;=\; \frac{1}{|D|} \sum_{(x,y) \in D} \mathbf{1}\{\hat{y}(x; P(S)) = y\}, \tag{1}$$

where $\hat{y}(x; P)$ is the evaluator's prediction under prompt $P$. Note that $S$ needs to have an order, since the prompt $P(S)$'s performance depends upon the order in which the few-shot examples occur.

For example $i \in [n]$, its Shapley value is the expected marginal contribution for all prompts randomly drawn from $[n]$:

$$\phi_i \;=\; \frac{1}{n! - (n-1)!} \sum_{\substack{\text{ordered subset } S \text{ of } [n] \\ i \in S}} \Big( v(S) - v(S \setminus \{i\}) \Big), \tag{2}$$

where $S \setminus \{i\}$ is the ordered subset with example $i$ removed. Note that $n! - (n-1)!$ is the number of ordered subsets of $[n]$ that include $i$.

**Remark 1** *We use Shapley values instead of a simpler and faster leave-one-out test because the former captures redundancy and complementarity of each example in the context of all other examples. This mitigates misattribution when examples overlap or interact—removal can look harmless if others duplicate it, or overly harmful if others depend on it. Empirically, the Shapley-based selector outperforms leave-one-out in our ablation (see Section 4, Table 2).*

**Monte-Carlo Approximation** Because summing over all permutations is infeasible, we approximate equation 2 with $K$ independently drawn random ordered subsets $\{S_k\}_{k=1}^K$ where $i \in S_k$ for all $k$:

$$\widehat{\phi}_i \;=\; \frac{1}{K} \sum_{k=1}^{K} \Big( v(S_k) - v(S_k \setminus \{i\}) \Big), \tag{3}$$

The pseudo-code for selecting the worst example is included in Appendix A.

**Replace/Drop/Keep decision.** Given the current set of in-context examples $[n]$ we determine the least helpful example $i^\star$ using the Shapley criterion:

$$i^\star \;=\; \arg\min_{i \in [n]} \widehat{\phi}_i. \tag{4}$$

For this step, the Example Improver proposes $m$ candidate examples $C = \{c_1, \ldots, c_m\}$ for potential appending to the current few shot example set. To decide whether to replace, keep, or drop the index $i^\star$, we compute the following scores:

$$r = \max_{c \in \mathcal{C}} v([n]\backslash\{i^\star\} \cup \{c\}) \qquad \text{(Replace)}$$

$$d = v([n]\backslash\{i^\star\}) \qquad \text{(Drop)}$$

$$k = v([n]) \qquad \text{(Keep)}$$

We select replace, keep or drop by checking whichever score is largest. When having ties, we prefer replace over drop over keep. The next prompt becomes $(N \setminus \{i^\star\}) \cup \{c^\star\}$ under REPLACE, $N \setminus \{i^\star\}$ under DROP, and $N$ under KEEP. This policy ensures we only adopt a modification when it does not underperform the best available alternative (drop or status quo).

**Remark 2** *Note that the Replace/Drop/Keep step could also be formulated directly using Shapley values. However, this would significantly increase the computational cost of each iteration, whereas our design prioritizes speed and efficiency.*

**Replay Buffer**  Our method relies on sampling a subset of the training data at each iteration. Consequently, newly crafted examples can overfit the current subset and fail to generalize to training subsets drawn in later iterations, since there is no mechanism enforcing that they also perform well on previously seen data. To mitigate this, after each iteration we store a small portion of the training data in a *replay buffer*. At the next iteration, this buffer is merged with the freshly sampled subset, which preserves accuracy across iterations by acting as a regularizer: newly crafted examples must also succeed on data sampled in prior iterations.

**Speed.**  To make our implementation fast, we employ the following techniques: We use a KV cache (Radford et al. (2019)) to avoid recomputing attention over already-processed tokens: Keys and values for the shared in-context prefix are cached once and then reused across (i) all tokens within a sequence and (ii) multiple evaluation queries that share this prefix. In addition, we rely on PagedAttention (Korthikanti et al. (2023)) to store the KV cache in paged memory chunks, which minimizes fragmentation and data movement while enabling efficient, contiguous access during decoding. Finally, we leverage continuous (token-level) batching (Yu et al. (2022)), in which the scheduler dynamically forms a new batch at each decoding step by admitting fresh requests and retiring completed ones, thereby overlapping prefill and decoding and maintaining high GPU utilization.

Pseudocode for PIAST can be found in Appendix A.

## 4 EXPERIMENTS

All experiments are conducted on single NVIDIA A100 GPU. We use the Qwen2.5-7B-Instruct model (Yang et al., 2024) as both the Example Generator, Improver and Prompt Evaluator for PIAST. Also all baselines are run using the same model, ensuring a fair comparison not distorted by LLM differences.

In this section, we present a series of experiments comparing PIAST against established baselines from the literature. Our evaluation spans four tasks: text classification, summarization, simplification and mathematical reasoning. In addition, we perform ablation studies to assess the contribution of individual hyperparameters, where each parameter is varied independently while keeping all others fixed. All experimental results, including both the main evaluations and ablation studies, are averaged over three runs for robustness. We use a single set of hyperparameters across all experiments. For hyperparameter values refer to Appendix F.

### 4.1 BASELINES

- **MI (Manual Instruction)** (Zhang et al., 2022): A set of prompts handcrafted and written by humans, aiming to improve task-specific performance.
- **NI (Natural Instruction)** (Mishra et al., 2021): Contains similarly to MI a set of human-written prompts for classification.
- **APE (Automatic Prompt Engineer)** (Zhou et al., 2022): Automatically generates multiple instruction candidates with an LLM and selects the most effective prompt based on downstream

performance, without further refinement during optimization. This method only rephrases instructions and does not generate few-shot examples.

- **APO (Automatic Prompt Optimization)** (Pryzant et al., 2023): Frames prompt tuning as a black-box optimization problem, refining prompts through an iterative feedback loop with beam search. Incorporate few-shot examples taken directly from the training dataset.
- **EvoPrompt** (Guo et al., 2023): Uses evolutionary strategies, selection, crossover, and mutation—to evolve a pool of discrete prompts and discover high-performing candidates. Similar to APE, only rephrases instructions and does not generate few-shot examples.
  - **DE (Differential Evolution)**: Explores the prompt space using differential evolution strategies.
  - **GA (Genetic Algorithm)**: Applies genetic operators such as selection, crossover, and mutation to progressively improve prompt quality.
- **PRL (Prompts from Reinforcement Learning)** (Batorski et al., 2025): Employs a reinforcement learning framework to automatically generate and optimize prompts. PRL also constructs few-shot examples that are not in the training set.
- **PIAST**: Our method as described in Section 3. The first two variants PIAST and PIAST(E) are used throughout experiments, while the (I) and (LOO) variants are ablations. All variants otherwise have the same hyperparameters.
  - **PIAST**: With medium runtime budget with limited access to the training set.
  - **PIAST (E)**: With extended runtime budget and accessing the full dataset.
  - **PIAST (I)**: Use only the initially generated examples, without the replace/keep/drop cycle. Notably this variant does not access the training set.
  - **PIAST (LOO)**: Replace Shapley value selection equation 2 by simple leave-one-out.

## 4.2 RESULTS

**Classification** We evaluate our method on a range of classification benchmarks covering sentiment, question, news, and subjectivity analysis:

- **Binary sentiment classification**: SST-2 (Socher et al., 2013), MR (Pang & Lee, 2005), and CR (Hu & Liu, 2004). The task is to classify each sentence as either `positive` or `negative`.
- **Multiclass sentiment classification**: SST-5 (Socher et al., 2013), extends sentiment analysis to five labels: `terrible`, `bad`, `okay`, `good`, and `great`.
- **Question classification**: TREC (Voorhees & Tice, 2000), questions categorization into one of six classes: `Description`, `Entity`, `Expression`, `Human`, `Location`, or `Number`.
- **News classification**: AG's News (Zhang et al., 2015), involves categorizing news articles into one of four topics: `World`, `Sports`, `Business`, or `Tech`.
- **Subjectivity classification**: SUBJ (Pang & Lee, 2004), the goal is to determine whether a sentence is `subjective` or `objective`.

Results are given in Table 2 and summarized in Figure 1, where also averaged runtimes and percentage data used are given. Detailed per-dataset runtimes are reported in Appendix B. As shown, PIAST consistently ranks among the top two methods in classification accuracy, while also being the fastest approach across all benchmarks. This demonstrates that efficient prompt construction not only reduces runtime but also maintains strong performance. Furthermore, the extended variant PIAST (E) yields additional improvements across all benchmarks, establishing new state–of–the–art results on AG's News and Subj. The example prompts are given in Appendix G.

**Simplification** We evaluate PIAST on the sentence simplification task using the ASSET dataset (Alva-Manchego et al., 2020). ASSET is a crowdsourced corpus specifically designed to evaluate simplification models across a range of rewriting operations, including lexical paraphrasing, sentence splitting, deletion, and reordering. Each original sentence is paired with multiple human-written simplifications.

For measuring simplification quality, we adopt the SARI metric (Xu et al., 2016), which compares the system output to both the original sentence and the reference simplifications. Results are presented in Table 3. As shown, PIAST achieves the highest SARI score for text simplification while requiring the least runtime. Furthermore, PIAST exhibits the lowest standard deviations across runs,

**Table 2:** Accuracy on classification tasks, averaged over three runs. Colours mark the best (red), second-best (orange) and third-best (yellow) numbers in each column; minor differences ($\leq 0.05$) are treated as ties. The right-most column shows the mean accuracy of each method across the seven datasets.

| Method / Dataset | SST-2 | CR | MR | SST-5 | AG's News | TREC | Subj | Avg |
|---|---|---|---|---|---|---|---|---|
| MI | 92.70 | 87.25 | 87.40 | 52.31 | 82.29 | 69.20 | 57.95 | 75.59 |
| NI | 95.77 | 91.50 | 90.85 | 51.90 | 83.43 | 66.60 | 68.10 | 78.31 |
| APO | $93.71_{\pm0.25}$ | $93.48_{\pm0.24}$ | $89.97_{\pm1.37}$ | $53.94_{\pm0.29}$ | $83.73_{\pm0.31}$ | $71.30_{\pm1.90}$ | $69.80_{\pm5.96}$ | 79.42 |
| APE | $91.23_{\pm0.66}$ | $92.87_{\pm0.02}$ | $89.90_{\pm0.94}$ | $49.37_{\pm5.66}$ | $82.58_{\pm1.20}$ | $77.07_{\pm1.61}$ | $73.92_{\pm1.39}$ | 79.56 |
| GA | $94.65_{\pm1.04}$ | $92.75_{\pm0.40}$ | $90.45_{\pm0.72}$ | $53.76_{\pm1.13}$ | $82.24_{\pm1.00}$ | $79.20_{\pm2.83}$ | $74.93_{\pm3.12}$ | 81.14 |
| DE | $93.29_{\pm0.34}$ | $93.38_{\pm0.19}$ | $89.98_{\pm0.24}$ | $55.25_{\pm0.37}$ | $82.18_{\pm1.04}$ | $76.47_{\pm0.38}$ | $73.08_{\pm4.95}$ | 80.52 |
| PRL | $96.32_{\pm0.04}$ | $92.83_{\pm0.24}$ | $91.27_{\pm0.05}$ | $56.21_{\pm0.15}$ | $84.36_{\pm0.08}$ | $77.07_{\pm2.36}$ | $76.90_{\pm0.95}$ | 82.14 |
| PIAST | $95.35_{\pm0.14}$ | $92.35_{\pm0.05}$ | $90.57_{\pm0.21}$ | $53.27_{\pm0.66}$ | $85.93_{\pm0.62}$ | $77.07_{\pm3.30}$ | $75.93_{\pm0.40}$ | 81.50 |
| PIAST (E) | $95.88_{\pm0.24}$ | $92.55_{\pm0.35}$ | $91.00_{\pm0.65}$ | $53.33_{\pm0.35}$ | $87.39_{\pm0.35}$ | $78.40_{\pm1.22}$ | $80.98_{\pm0.67}$ | 82.79 |
| PIAST (I) | $95.04_{\pm0.18}$ | $91.53_{\pm0.65}$ | $90.43_{\pm0.21}$ | $49.79_{\pm1.05}$ | $85.38_{\pm0.20}$ | $74.33_{\pm4.77}$ | $59.52_{\pm2.29}$ | 78.00 |
| PIAST (LOO) | $95.70_{\pm0.31}$ | $92.15_{\pm0.15}$ | $90.42_{\pm0.28}$ | $53.18_{\pm1.40}$ | $86.43_{\pm0.72}$ | $75.87_{\pm2.37}$ | $69.73_{\pm3.68}$ | 80.50 |

highlighting its stability and robustness. With additional computational time and data, PIAST (E) attains an even higher score on this benchmark. The example prompt is given in Appendix H.

**Summarization** We evaluate PIAST on the task of text summarization, where the goal is to extract and condense the most salient information from an input passage preserving key content while omitting redundant or irrelevant details.

Our experiments are conducted on the SAMSUM dataset (Gliwa et al., 2019), a collection of English-language chat dialogues designed to resemble real-life messenger conversations.

For evaluation, we report scores using the widely adopted ROUGE metrics (Lin, 2004). ROUGE-1 measures unigram similarity, reflecting context coverage. ROUGE-2 measures bigram similarity, reflecting local coherence and phrasing. ROUGE-L measures longest common subsequence, measuring fluency and structural alignment.

**Table 3:** Simplification task results.

| Method | SARI ($\uparrow$) | Time [m] ($\downarrow$) |
|---|---|---|
| MI | 43.77 | – |
| APE | $45.33_{\pm0.83}$ | $35.69_{\pm0.20}$ |
| GA | $46.25_{\pm0.47}$ | $39.60_{\pm0.63}$ |
| DE | $45.79_{\pm0.35}$ | $52.77_{\pm1.12}$ |
| PRL | $52.26_{\pm3.51}$ | $2880.00_{\pm0.00}$ |
| PIAST | $54.52_{\pm0.07}$ | $18.14_{\pm0.42}$ |
| PIAST (E) | $55.06_{\pm0.26}$ | $389.78_{\pm113.17}$ |

**Table 4:** Text summarization results with ROUGE scores and runtime (minutes).

| Method | ROUGE-1 | ROUGE-2 | ROUGE-L | Time [m] |
|---|---|---|---|---|
| MI | 32.76 | 10.39 | 28.97 | – |
| APE | $37.12_{\pm2.02}$ | $12.97_{\pm0.74}$ | $33.32_{\pm1.68}$ | $60.07_{\pm0.27}$ |
| GA | $39.69_{\pm1.76}$ | $14.47_{\pm1.00}$ | $35.84_{\pm1.63}$ | $89.31_{\pm3.08}$ |
| DE | $33.91_{\pm4.04}$ | $12.53_{\pm1.47}$ | $31.05_{\pm3.79}$ | $76.89_{\pm1.34}$ |
| PRL | $42.47_{\pm0.83}$ | $16.17_{\pm0.24}$ | $37.73_{\pm0.36}$ | $2880.00_{\pm0.00}$ |
| PIAST | $41.13_{\pm0.67}$ | $16.07_{\pm0.76}$ | $36.74_{\pm0.48}$ | $34.48_{\pm0.27}$ |
| PIAST (E) | $42.13_{\pm0.27}$ | $16.83_{\pm0.3}$ | $37.37_{\pm0.25}$ | $737.00_{\pm108.31}$ |

Results are summarized in Table 4. As shown, PIAST is the fastest method while consistently ranking second across all evaluation metrics. An interesting observation is that, although PRL is capable of generating examples, it does not utilize any for the summarization task. Instead, PRL merely rephrases the manual prompt. The authors of PRL argue that summarization is not particularly suitable for example-based prompting. While we find that incorporating examples can indeed enhance performance, the improvements do not reach the level achieved by PRL. Moreover, we observe that PIAST (E) further improves the results, attaining state-of-the-art performance on the ROUGE-2 metric as well as on the average of ROUGE-1, ROUGE-2, and ROUGE-L. The example prompt of PIAST is provided in Appendix I.

**GSM8K** We evaluate our approach on the GSM8K dataset (Cobbe et al., 2021), a benchmark that demands explicit, multi-step arithmetic reasoning where answers are unconstrained integers—making robust prompt design especially critical. Prior work has shown that such reasoning performance is highly sensitive to the choice of exemplars (Wei et al., 2022). As summarized in Table 5, methods that only adjust the base prompt (APE, GA, DE) yield modest gains over the MI baseline, whereas approaches that incorporate few-shot examples (PRL, PIAST) achieve substan-

tially stronger accuracy. Notably, PIAST and PIAST (E) obtain the best and second to best results, respectively, with PIAST also being the fastest among the top-performing methods. These findings indicate that PIAST is both competitive and efficient for reasoning intensive tasks like GSM8K.

### 4.3 ABLATIONS

Table 5: GSM8K Results.

| Method | Acc. | Time [m] |
|---|---|---|
| MI | 78.20 | – |
| APE | $83.43_{\pm1.98}$ | $180.81_{\pm2.66}$ |
| GA | $81.62_{\pm1.38}$ | $191.96_{\pm1.11}$ |
| DE | $79.52_{\pm0.45}$ | $252.57_{\pm3.59}$ |
| PRL | $86.15_{\pm0.55}$ | $2880.00_{\pm0.00}$ |
| PIAST | $91.65_{\pm0.31}$ | $80.26_{\pm2.95}$ |
| PIAST (E) | $92.12_{\pm0.12}$ | $1598.34_{\pm234.54}$ |

**Ablation Study: Cross-Model Robustness** We assess how well prompts learned by PIAST transfer across models in two settings. **(i) Cross-model inference.** We train prompts with Qwen2.5-7B-Instruct and then evaluate *the same prompts* using Mistral-7B-Instruct-v0.2 Jiang et al. (2023). As shown in Table 6, PIAST attains the strongest transfer on SUBJ, edging out PRL and APO; APE and GA are roughly on par with the manual instruction baseline. Notably, PIAST (E) exhibits a sizable accuracy drop, which we attribute to overfitting to the source evaluator due to its substantially larger improvement iteration budget. These results suggest that, when portability matters, PIAST offers the best cross-model robustness. **(ii) Component swaps.** We next vary which model plays each role, swapping Qwen and Mistral between the (Example Proposer & Improver) and the Prompt Evaluator. Table 7 shows that accuracy remains comparable across configurations, indicating that PIAST is not overly sensitive to a particular model pairing.

Table 6: Cross-model inference on SUBJ: prompts trained with Qwen2.5-7B-Instruct, evaluated with Mistral-7B-Instruct-v0.2.

| **Method** | **Acc.** |
|---|---|
| MI | 60.30 |
| APE | $60.77_{\pm1.08}$ |
| APO | $69.53_{\pm1.33}$ |
| GA | $60.68_{\pm1.60}$ |
| DE | $64.10_{\pm2.20}$ |
| PRL | $70.73_{\pm3.81}$ |
| PIAST | $72.87_{\pm4.16}$ |
| PIAST (E) | $68.75_{\pm3.01}$ |

**Ablation Study: Influence of the Replace/Drop/Keep Optimization** In this experiment, we evaluate PIAST without the optimization loop, i.e., the model is tested directly on the proposed initial examples. We include this variant as a baseline, denoted PIAST (I), in Table 2. As shown, for many benchmarks the initial examples already yield strong performance, in some cases even surpassing algorithms that employ optimization loops. Nevertheless, we consistently observe that incorporating our optimization loop further improves the results. For certain tasks, such as binary sentiment classification (e.g., SST-2 or MR), the improvement is marginal. We attribute this to the fact that the initial examples are already highly effective, as the underlying LLM has a strong capability to distinguish between positive and negative samples.

Interestingly, in the subjectivity dataset PIAST without the optimization loop performs poorly, achieving results comparable to those of the Manual Instruction baseline. However, after applying the optimization loop, performance improves significantly by 16.41% and by an additional 5.05% when using PIAST(E), showing that on more difficult tasks where in-context example selection is non-trivial our optimization loop can help a lot.

Table 7: Component swaps on SUBJ: accuracy when exchanging the Example Proposer (P) & Improver (I) and the Prompt Evaluator between Qwen and Mistral.

| **P & I** | **Eval** | **Acc.** |
|---|---|---|
| Qwen | Mistral | $75.93_{\pm3.14}$ |
| Mistral | Qwen | $72.52_{\pm1.71}$ |
| Mistral | Mistral | $74.93_{\pm2.54}$ |

**Ablation Study: Influence of #Replace/Drop/Keep Iterations** We investigate how PIAST scales with an increasing number of crafting iterations, and compare it against baselines from the literature. To this end, we run PIAST with 10, 15, 30, 50, 100, and 150 crafting iterations, and report the results in Figure 3.

We observe a clear trend: increasing the number of crafting iterations consistently improves accuracy, albeit at the cost of higher runtime. This highlights an appealing property of PIAST: its performance can be effectively scaled by allocating more computation time by increasing the number of crafting iterations. Moreover, the plot clearly shows that PIAST has anytime performance superior to the baselines.

**Ablation: Leave–One–Out vs. Shapley for worst–example selection**    To test whether a simpler procedure can replace our Shapley–value selection, we evaluate a *leave–one–out* (LOO) heuristic for identifying the worst (most harmful) in–context example. Given $n$ examples $E = \{e_i\}_{i=1}^n$, LOO removes each example once and measures performance on the validation split:

$$i^\star = \arg \max_{i \in \{1, \ldots, N\}} v(E \setminus \{e_i\}).$$

In words, LLO chooses the example whose removal yields the highest accuracy drop. removing a strongly useful example would decrease it.

We run this ablation on all classification tasks using the same hyperparameters as PIAST and report results in Table 2. Across several benchmarks LOO attains results comparable to our full method, but on SUBJ there is a clear gap between PIAST and PIAST (LOO). We hypothesize that, for many classification tasks, the initial pool already contains mostly good examples, so LOO can make small, beneficial adjustments. In contrast, SUBJ is more sensitive to initialization (see PIAST (I)), and the Shapley–based selection is notably more robust in such settings, where performance depends heavily on which examples are initially presented.

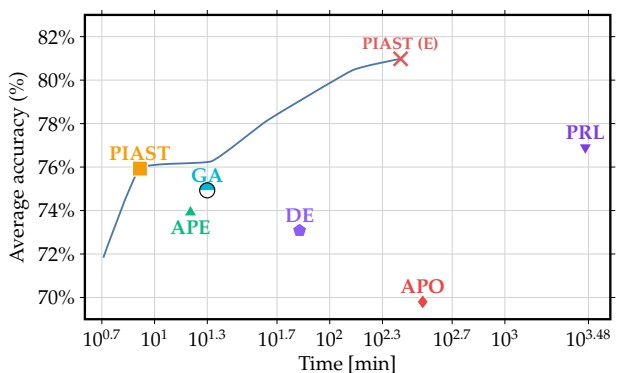

**Figure 3:** Scaling of PIAST on SUBJ compared to other baselines while increasing the number of improvement iterations.

**Ablation: Number of Shapley Permutations**    We study the sensitivity of PIAST to the number of Monte–Carlo permutations $K$ used in the Shapley-value estimator equation 3. As shown in Table 8, increasing $K$ beyond 3 yields only marginal accuracy gains (at most 0.39 points from $K = 3$ to $K = 50$) while substantially increasing runtime. Using $K = 1$ underperforms $K = 3$ by 2.25 points, indicating that a small amount of averaging is beneficial. We therefore adopt $K = 3$ as our default, which offers a strong performance/speed trade-off.

## 5   CONCLUSIONS & LIMITATIONS

Constructing prompts with in-context examples is key for performance in automatic prompting. Our method PIAST shows that this can be done fast and data-lean.

It is an interesting open question how PIAST would perform if the underlying LLM has little task specific competence to suggest good examples. We also believe that, while arguably in-context examples are more impactful overall, combining PIAST with prompt rewriting might yield some additional benefits. Last, PIAST is in a loose sense similar to

**Table 8:** Effect of the number of Shapley permutations on subjectivity.

| P | Acc. | Time [m] |
|---|---|---|
| 1 | $73.68_{\pm 1.45}$ | $4.61_{\pm 0.04}$ |
| 3 | $75.93_{\pm 0.40}$ | $8.25_{\pm 0.27}$ |
| 10 | $76.02_{\pm 1.08}$ | $20.88_{\pm 0.17}$ |
| 50 | $76.32_{\pm 1.41}$ | $83.90_{\pm 1.82}$ |

a local neighborhood search heuristics in optimization, where our local neighborhood are the current examples and changing neighborhoods can be done via dropping or replacing examples. It would be an interesting problem to see whether other ideas from primal heuristics could be used to search the combinatorial space of few-shot examples effectively.

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

# A PSEUDOCODES

We present concise pseudocodes for our method and its Shapley-driven oracle. Algorithm 1 orchestrates the full crafting loop: starting from $k$ initial examples, it performs $I$ rounds that evaluate on a small subsample (size $s$) augmented with a replay buffer (size $r$). In each round, MONTECARLOSHAPLEYWORST identifies the least helpful example using $P$ random permutations, the improver proposes $m$ candidates, and a conservative REPLACE/DROP/KEEP rule updates the set only when accuracy does not regress. Algorithm 2 details the Shapley routine with memoized coalition values $v(S)$ and permutation-averaged marginal contributions, returning the index with the smallest estimated value.

---

**Algorithm 1** PIAST

---

**Require:**
    Dataset $\mathcal{D}$
    Example Proposer $M_{\text{prop}}$
    Prompt Evaluator $M_{\text{eval}}$
    Example Improver $M_{\text{impr}}$
    $k$ (Initial examples)
    $I$ (Number of craft iterations)
    $m$ (Number of refine candidates)
    $s$ (Size of subdataset)
    $r$ (Replay Size)
    $P$ (Number of Shapley Permutations)
**Ensure:** Crafted example set $E^{\star}$

1: $E \leftarrow$ PROPOSEINITIALEXAMPLES$(M_{\text{prop}}, k)$               $\triangleright |E| = k$
2: $R \leftarrow \varnothing$                  $\triangleright$ Replay buffer
3: **for** $t = 0, 1, \ldots, I - 1$ **do**
4:      $D_t \leftarrow$ SUBSAMPLE$(\mathcal{D}_{\text{infer}}, s)$
5:      $\tilde{D}_t \leftarrow D_t \cup R$               $\triangleright$ Union with replay
6:      $a_{\text{base}} \leftarrow$ EVALACC$(M_{\text{eval}}, E, \tilde{D}_t)$
7:      $i^{\star} \leftarrow$ MONTECARLOSHAPLEYWORST$(E, \tilde{D}_t, M_{\text{eval}}, P)$
8:      $E_{\setminus i^{\star}} \leftarrow E \setminus \{e_{i^{\star}}\}$
9:      $a_{\text{drop}} \leftarrow$ EVALACC$(M_{\text{eval}}, E_{\setminus i^{\star}}, \tilde{D}_t)$
10:      $C \leftarrow$ GENERATECANDIDATES$(M_{\text{impr}}, E_{\setminus i^{\star}}, m)$
11:      $(c^{\text{best}}, a_{\text{best}}) \leftarrow \arg\max\limits_{c \in C}$ EVALACC$(M_{\text{eval}}, E_{\setminus i^{\star}} \cup \{c\}, \tilde{D}_t)$
                                                $\triangleright$ **Decision**: REPLACE vs DROP vs KEEP
12:      **if** $a_{\text{best}} \geq a_{\text{drop}}$ **and** $a_{\text{best}} \geq a_{\text{base}}$ **then**
13:          $E \leftarrow E_{\setminus i^{\star}} \cup \{c^{\text{best}}\}$          $\triangleright$ REPLACE
14:      **else if** $a_{\text{drop}} \geq a_{\text{base}}$ **and** $|E| > 1$ **then**
15:          $E \leftarrow E_{\setminus i^{\star}}$             $\triangleright$ DROP
16:      **else**
17:          $E \leftarrow E$                 $\triangleright$ KEEP
18:      **end if**
19:      $R \leftarrow R \cup$ SAMPLEREPLAY$(D_t, r)$
20: **end for**
21: $E^{\star} \leftarrow E$
22: **return** $E^{\star}$

---

---

**Algorithm 2** MONTECARLOSHAPLEYWORST

---

**Require:**
    Example set $E = \{e_1, \ldots, e_n\}$
    Dataset subset $\tilde{D}$
    Prompt Evaluator $M_{\text{eval}}$
    $P$ (Number of Shapley permutations)
**Ensure:** Worst index $i^\star$

1:  $\mathcal{V} \leftarrow \varnothing$
2:  For each $i \in [n]$, set list $\Delta_i \leftarrow [\,]$
3:  Define $v(S) \leftarrow \text{EVALACC}(M_{\text{eval}}, \{e_j : j \in S\}, \tilde{D})$
4:  $\mathcal{V}[\varnothing] \leftarrow v(\varnothing)$
5:  **for** $p = 1, 2, \ldots, P$ **do**
6:     $\pi \leftarrow$ a random permutation of $[n]$
7:     $S \leftarrow \varnothing$;  $v_{\text{prev}} \leftarrow \mathcal{V}[\varnothing]$
8:     **for** $j = 1, 2, \ldots, n$ **do**
9:         $i \leftarrow \pi_j$;  $S' \leftarrow S \cup \{i\}$
10:      **if** $S' \notin \mathcal{V}$ **then**
11:         $\mathcal{V}[S'] \leftarrow v(S')$
12:      **end if**
13:      $v_{\text{new}} \leftarrow \mathcal{V}[S']$
14:      Append $(v_{\text{new}} - v_{\text{prev}})$ to $\Delta_i$               $\triangleright$ Marginal contribution of $e_i$
15:      $S \leftarrow S'$;  $v_{\text{prev}} \leftarrow v_{\text{new}}$
16:     **end for**
17: **end for**
18: **for** $i = 1, 2, \ldots, n$ **do**
19:     $\phi_i \leftarrow \begin{cases} \frac{1}{|\Delta_i|} \sum\limits_{d \in \Delta_i} d, & |\Delta_i| > 0 \\ 0, & \text{otherwise} \end{cases}$
20: **end for**
21: $i^\star \leftarrow \arg\min\limits_{i \in [n]} \phi_i$
22: **return** $i^\star$

---

## B   RUNTIME ANALYSIS FOR CLASSIFICATION BENCHMARKS

In this section, we report the average runtime (in minutes) across different methods for each classification benchmark. Table 9 summarizes the results, including mean and standard deviation values.

**Table 9:** Average runtime in minutes (mean $\pm$ SD) for each classification benchmark and method. The last row reports the overall average across all APO runs.

| Dataset | APE | APO | DE | GA | PRL | PIAST | PIAST (E) |
|---|---|---|---|---|---|---|---|
| sst2 | $62.85_{\pm 1.96}$ | $376.82_{\pm 0.00}$ | $62.79_{\pm 2.54}$ | $20.21_{\pm 2.84}$ | $2880.00_{\pm 0.00}$ | $7.49_{\pm 0.14}$ | $221.48_{\pm 8.41}$ |
| cr | $16.09_{\pm 5.29}$ | $302.22_{\pm 47.40}$ | $65.06_{\pm 0.96}$ | $18.75_{\pm 4.37}$ | $2880.00_{\pm 0.00}$ | $7.49_{\pm 0.20}$ | $181.06_{\pm 8.67}$ |
| mr | $4.60_{\pm 0.45}$ | $342.03_{\pm 15.04}$ | $62.71_{\pm 1.36}$ | $28.84_{\pm 9.18}$ | $2880.00_{\pm 26.19}$ | $7.64_{\pm 0.11}$ | $210.40_{\pm 20.84}$ |
| sst5 | $5.99_{\pm 1.06}$ | $430.08_{\pm 92.80}$ | $62.09_{\pm 1.88}$ | $20.68_{\pm 1.82}$ | $2880.00_{\pm 0.00}$ | $7.63_{\pm 0.11}$ | $155.55_{\pm 0.35}$ |
| agnews | $7.33_{\pm 2.43}$ | $241.19_{\pm 28.07}$ | $65.16_{\pm 3.59}$ | $18.08_{\pm 2.54}$ | $2880.00_{\pm 0.00}$ | $8.38_{\pm 0.16}$ | $210.97_{\pm 38.85}$ |
| trec | $3.94_{\pm 0.74}$ | $256.34_{\pm 17.55}$ | $66.01_{\pm 3.25}$ | $22.24_{\pm 6.46}$ | $2880.00_{\pm 0.00}$ | $6.63_{\pm 0.19}$ | $146.19_{\pm 28.15}$ |
| subj | $16.03_{\pm 2.63}$ | $339.44_{\pm 42.38}$ | $67.28_{\pm 0.37}$ | $19.98_{\pm 2.33}$ | $2880.00_{\pm 0.00}$ | $8.25_{\pm 0.27}$ | $253.83_{\pm 8.07}$ |

## C   HYPERPARAMETER STUDY

We study the impact of the number of Shapley permutations, number of refine and initial example candidates for performance of PIAST.

**Table 10:** Effect of the number of proposed candidates on the subjectivity task.

| m | Acc. | Time (min) |
|---|---|---|
| 5 | $70.57_{\pm 2.33}$ | $7.11_{\pm 0.15}$ |
| 10 | $75.93_{\pm 0.07}$ | $8.25_{\pm 0.27}$ |
| 30 | $76.15_{\pm 0.48}$ | $11.54_{\pm 0.05}$ |
| 50 | $76.33_{\pm 1.53}$ | $12.99_{\pm 0.32}$ |

**Hyperparameter Study: Influence of Number of Refine Candidates** In this experiment, we analyze how the number of candidate examples $m$ proposed by the example improver influences the final performance. The results are summarized in Table 10.

We observe that using too few candidates ($m = 5$) leads to a significant drop in accuracy, which can be mitigated by increasing the number of candidates to $m = 10$. For larger values $m = 30$ and $m = 50$, accuracy does not improve substantially, suggesting that an initial pool of $m = 10$ candidates combined with the replace/drop/keep iteration is sufficient to achieve strong performance. Therefore, in all subsequent experiments we adopt $m = 10$ as the default setting.

**Hyperparameter Study: Influence of Number of Initial Examples** In this experiment, we investigate how performance on the subjectivity task varies with the number of initial examples. The results are reported in Table 11. The number of initial examples does not exhibit a clear monotonic trend (e.g., "the more, the better"). Using too few initial examples may lead to underfitting, as they fail to provide sufficient insight into the task. Conversely, using too many examples can introduce excessive noise, making it more difficult for the evaluator to identify the weakest examples within a large pool. Based on this analysis, we select 16 examples as the most robust choice for our experiments.

**Table 11:** Effect of the number of initial examples on the subjectivity task.

| k | Acc. | Time [m] |
|---|---|---|
| 4 | $71.77_{\pm 0.42}$ | $2.88_{\pm 0.12}$ |
| 8 | $74.20_{\pm 1.66}$ | $4.63_{\pm 0.14}$ |
| 16 | $75.93_{\pm 0.40}$ | $8.25_{\pm 0.27}$ |
| 32 | $74.77_{\pm 0.48}$ | $16.26_{\pm 0.26}$ |

# D PROMPT FOR EXAMPLE GENERATOR

In this section we give the prompt for the prompt evaluator for binary sentimental analysis task. For other tasks, we prompt is analogical.

**Example Proposer prompt for CR dataset**

You are a data generator that writes high-quality in-context learning examples for *binary sentiment* on short movie-review style snippets. Create exactly {NUM_EXAMPLES} training examples in THIS STRICT format only:

Example1:
Sentence: ""<text>""
Label: {LABEL}

Example2:
Sentence: ""<text>""
Label: {LABEL}

...

Example{NUM_EXAMPLES}:
Sentence: ""<text>""
Label: {LABEL}

Diversity plan (MUST FOLLOW):
{DIVERSITY_PLAN}

Rules:
- Each example's "Sentence" must contain exactly the number of sentences specified above (1–3).
- Keep sentences concise: typically 3–14 words each. Across the set, include at least one very short ($\leq 5$ words) and one longer (10–14 words).
- Use only ASCII characters. Do NOT include double quotes inside the text.
- Use exactly ONE 'Sentence:' line per example; if multiple sentences are needed, put them inside the same quotes separated by a space.
- Make the writing naturally match the requested label in the everyday sense of the word.
- Do NOT mention the label or talk about labels in the text (no meta commentary).
- No Markdown/code fences.
- Output ONLY the examples in the exact format above; no extra text.

## E  PROMPT FOR PROMPT IMPROVER

**Example Improver prompt for CR dataset**

You are improving in-context examples for sentiment classification. Generate replacements that diversify length (1–3 sentences) and topic, avoid paraphrasing, and help the task.

You are given the CURRENT examples (do not repeat or paraphrase them):
{CURRENT_EXAMPLES}

Now create exactly {NUM_CANDIDATES} NEW examples in THIS STRICT format:
Example1:
Sentence: ""<text>""
Label: positive|negative

Example2:
Sentence: ""<text>""
Label: positive|negative

...

Example{NUM_CANDIDATES}:
Sentence: ""<text>""
Label: positive|negative

Diversity plan (MUST FOLLOW):
{DIVERSITY_PLAN}

Rules:
- Use exactly ONE 'Sentence:' line per example. If multiple sentences are needed, put them INSIDE the same quotes separated by a space.
- Each example must have exactly the number of sentences specified in the plan above (1–3).
- Keep sentences concise: typically 3–14 words each. Across the set, include very short ($\leq 5$ words) and longer (10–14 words).
- ASCII only. Do NOT include double quotes inside the text.
- Make topics clearly different from the given examples and from each other; avoid near-duplicates or paraphrases.
- Prefer balancing labels; if unsure, choose the minority label: {MINORITY_LABEL}.
- Do NOT wrap output in Markdown/code fences.
- Output ONLY the examples in the exact format above; no extra text.

## F  Hyperparameters

In this section, we present the exact hyperparameters used across all tasks. A single set of hyperparameters was applied consistently across all tasks, and their values are summarized in Table 12.

| Hyperparameter | Value |
| --- | --- |
| $k$ | 16 |
| $s$ | 70 |
| $I$ | 15 |
| $m$ | 10 |
| $r$ | 5 |
| $P$ | 3 |

**Table 12:** Hyperparameters used across all tasks. The notation is consistent with the pseudocode provided in Appendix A.

The hyperparameters for PIAST (E) are identical to those of PIAST, except that we increase the number of iterations $I$ to 150. This adjustment provides the most effective way to scale the performance of PIAST, as demonstrated in our ablation studies (see Section 4.3).

## G  Classification Prompts

In this section, we describe the most effective prompts for PIAST on classification tasks. The base prompts are taken from Guo et al. (2023), and the examples are produced by our method.

> **SST2**
>
> Please perform Sentiment Classification task. Given the sentence, assign a sentiment label from ['negative', 'positive']. Return label only without any other text.
> Example1: Sentence: "The film maintains a steady pace. No dull moments. Engaging from beginning to end." Label: positive
> Example2: Sentence: "The set design is detailed. Costumes match the era perfectly. Attention to historical accuracy is evident." Label: positive
> Example3: Sentence: "The lead actor delivers a compelling performance." Label: positive
> Example4: Sentence: "The film maintains a brisk pace from start to finish. No lulls or wasted moments, just continuous action and intrigue." Label: positive
> Example5: Sentence: "The soundtrack is loud and distracting, overshadowing the dialogue." Label: negative
> Example6: Sentence: "The editing is seamless, keeping the pace tight. Transitions between scenes are smooth and impactful." Label: positive
> Example7: Sentence: "The lead actor's performance is powerful and moving." Label: positive
> Example8: Sentence: "The cinematography is breathtaking, capturing the essence of the story. The use of light and color enhances every scene." Label: positive
> Example9: Sentence: "The lead actress's performance is powerful." Label: positive
> Example10: Sentence: "The direction was confusing and lacked focus." Label: negative
> Example11: Sentence: "The lead actress shines in every scene." Label: positive
> Example12: Sentence: "The dialogue felt forced and unnatural." Label: negative
> Example13: Sentence: "The visual effects were poorly done. The CGI looked cheap and unrealistic. It ruined the immersion." Label: negative
> Example14: Sentence: "The director masterfully guides the narrative." Label: positive
> Example15: Sentence: "The lead actress's portrayal is emotionally resonant." Label: positive

> **CR**
>
> Please perform Sentiment Classification task. Given the sentence, assign a sentiment label from ['negative', 'positive']. Return label only without any other text.

Example1: Sentence: "The cinematography captured the essence of the setting, with stunning visuals that added depth to the story." Label: positive
Example2: Sentence: "The editing was precise, enhancing the flow of the story. However, some transitions felt abrupt and jarring." Label: negative
Example3: Sentence: "The actors delivered powerful performances, bringing depth to their roles." Label: positive
Example4: Sentence: "The screenplay was weak, with dialogue that felt forced and unnatural. Characters lacked development, making their actions confusing. The dialogue felt stiff, with lines that didn't flow naturally. This made the scenes less engaging." Label: negative
Example5: Sentence: "The first act was slow and" Label: positive
Example6: Sentence: "The editing was seamless, enhancing the flow of the story. Cuts were precise, keeping the pacing tight." Label: positive
Example7: Sentence: "The film started slowly but picked up in the middle." Label: negative
Example8: Sentence: "The director's vision was clear but the execution was lacking. Scenes felt disjointed, and the overall story was confusing." Label: negative
Example9: Sentence: "The editing was choppy and disjointed." Label: negative
Example10: Sentence: "The director's vision was clear, but the actors seemed uncomfortable on camera." Label: negative
Example11: Sentence: "The screenplay felt rushed, with dialogue that seemed out of place. Characters had little to no development, making their motivations unclear. The plot relied too heavily on clichés, lacking originality." Label: negative
Example12: Sentence: "The director's vision was clear and inspiring. However, the final cut felt rushed and incomplete." Label: negative
Example13: Sentence: "The soundtrack was inappropriate, detracting from the mood of the scenes." Label: negative
Example14: Sentence: "The director skillfully guided the ensemble cast." Label: positive
Example15: Sentence: "The screenplay felt rushed, with dialogue that seemed out of place. Characters had little to no development, making their motivations unclear. The plot relied too heavily on clichés, lacking originality." Label: negative

## MR

Please perform Sentiment Classification task. Given the sentence, assign a sentiment label from ['negative', 'positive']. Return label only without any other text.
Example1: Sentence: "The editing was choppy, disrupting the flow of the narrative. Scenes felt disjointed, and the timing was off." Label: negative
Example2: Sentence: "The lead actress delivered a nuanced and emotionally rich performance." Label: positive
Example3: Sentence: "The movie started slowly but picked up momentum. The second act was particularly well-paced, maintaining tension." Label: positive
Example4: Sentence: "The lead actor's performance was powerful and moving." Label: positive
Example5: Sentence: "The lead actress captivated the audience with her portrayal." Label: positive
Example6: Sentence: "The soundtrack was overbearing and distracting." Label: negative
Example7: Sentence: "The soundtrack added a melancholic tone that complemented the film's somber mood." Label: positive
Example8: Sentence: "The actor's portrayal was compelling and emotionally resonant." Label: positive
Example9: Sentence: "The editing was choppy and disjointed." Label: negative
Example10: Sentence: "The lead actor brought depth to the role." Label: positive
Example11: Sentence: "The lead actor's portrayal was gripping and heartfelt." Label: positive
Example12: Sentence: "The soundtrack added a perfect touch, enhancing the film's dramatic moments." Label: positive
Example13: Sentence: "The visual effects were poorly done and noticeable." Label: negative
Example14: Sentence: "The director's vision was unclear, leading to a disjointed narrative. The characters felt underdeveloped." Label: negative
Example15: Sentence: "The acting was wooden and unconvincing." Label: negative

## SST5

Please perform Sentiment Classification task. Given the sentence, assign a sentiment label from ['terrible', 'bad', 'okay', 'good', 'great']. Return label only without any other text.

Example1: Sentence: "The screenplay was predictable. The dialogue lacked depth, feeling forced. Characters felt flat and uninteresting." Label: bad

Example2: Sentence: "The story lacked coherence and felt rushed. The plot had too many loose ends and felt unsatisfying." Label: bad

Example3: Sentence: "The screenplay was cliché and predictable. The dialogue lacked depth and felt forced." Label: bad

Example4: Sentence: "The story was predictable with a weak ending. It lacked the twists needed for a compelling narrative." Label: okay

Example5: Sentence: "The screenplay was clever and witty. The dialogue was sharp and well-paced. It captured the essence of the characters perfectly." Label: great

Example6: Sentence: "Direction felt disjointed and confusing." Label: terrible

Example7: Sentence: "Story lacked coherence and felt rushed." Label: terrible

Example8: Sentence: "Dialogue was forced and awkward, breaking the mood." Label: terrible

Example9: Sentence: "Direction kept the pace just right; not too slow or fast." Label: good

Example10: Sentence: "The dialogue was cliché and predictable. The script failed to deliver any surprises." Label: okay

Example11: Sentence: "The director skillfully balanced the dramatic and comedic elements. The pacing was just right, keeping the audience engaged. The visual storytelling was top-notch." Label: great

Example12: Sentence: "Dialogue was sharp and added depth to the characters." Label: good

Example13: Sentence: "The acting was solid but not memorable. The supporting cast added some depth to the film." Label: okay

Example14: Sentence: "Acting was wooden and unconvincing." Label: terrible

Example15: Sentence: "Great performances by all; especially the lead actor." Label: good

Example16: Sentence: "The acting was wooden and unconvincing." Label: bad

Example17: Sentence: "The direction felt disjointed and confusing." Label: bad

Example18: Sentence: "The direction was uninspired. The pacing felt slow and the camera work was basic." Label: okay

**AG's News**

Please perform News Classification task. Given the news item, assign a label from ['World', 'Sports', 'Business', 'Tech']. Return label only without any other text

Example1: Sentence: "Upcoming elections will focus on healthcare reform and immigration policies; debates intensify." Label: World

Example2: Sentence: "NASA launches Mars rover to study planet's geology." Label: Tech

Example3: Sentence: "Supreme Court rules on new labor laws; impact on businesses discussed." Label: Business

Example4: Sentence: "Elections this year will focus on healthcare and education reform." Label: World

Example5: Sentence: "Telemedicine platforms see surge in usage during pandemic." Label: Tech

Example6: Sentence: "US and China engage in summit talks to discuss trade and climate issues; tensions remain high." Label: Business

Example7: Sentence: "Vaccination drive reaches remote villages successfully." Label: World

Example8: Sentence: "Diplomatic talks on climate change continue with mixed progress." Label: World

Example9: Sentence: "Bombing kills dozens in city center." Label: World

Example10: Sentence: "Upcoming elections will focus on healthcare and economic reforms; debates intensify as candidates present their plans." Label: World

Example11: Sentence: "Security forces respond to a terrorist attack in the city center; multiple casualties reported. Emergency services work to contain the situation." Label: World

Example12: Sentence: "Diplomatic talks on trade agreements between Asia-Pacific nations continue." Label: World

Example13: Sentence: "Upcoming elections will focus on healthcare and economic reforms; debates intensify as candidates present their plans. Voters express concerns about rising costs." Label: World

Example14: Sentence: "Diplomatic talks between nations on climate change progress despite initial disagreements." Label: World

**TREC**

Please perform Question Classification task. Given the question, assign a label from ['Description', 'Entity', 'Expression', 'Human', 'Location', 'Number']. Return label only without any other text.

Example1: Sentence: "Calculus is a branch of mathematics that deals with rates of change and slopes of curves. It includes differential and integral calculus. The fundamental theorem of calculus links these two concepts." Label: Description

Example2: Sentence: "Who wrote the screenplay for the movie where the main character delivers a famous monologue about the American Dream?" Label: Human
Example3: Sentence: "The human genome consists of all the genetic information in a human cell. It is composed of approximately 3 billion base pairs." Label: Number
Example4: Sentence: "The Magna Carta, signed in 1215, was a landmark document in English history." Label: Description
Example5: Sentence: "The director chose to shoot the film in black and white to evoke a sense of nostalgia." Label: Description
Example6: Sentence: "Calculus involves the study of limits, derivatives, integrals, and infinite series. It is essential for understanding changes in various quantities." Label: Description
Example7: Sentence: "She delivered the line with such conviction it seemed real. The audience was moved." Label: Expression
Example8: Sentence: "Which director is known for their innovative use of camera angles in films?" Label: Human
Example9: Sentence: "Meryl Streep has performed in many famous plays." Label: Location
Example10: Sentence: "He directed the actors to bring out the raw emotion in their performances. The result was powerful." Label: Expression
Example11: Sentence: "The screenplay's dialogue was sharp and witty, setting the tone for the entire film." Label: Expression
Example12: Sentence: "Newton's laws of motion describe how objects move under the influence of forces." Label: Description
Example13: Sentence: "The screenplay features complex dialogue that drives the characters' motivations and relationships." Label: Description
Example14: Sentence: "Recent studies show that vitamin C is crucial for the immune system. It helps in fighting infections." Label: Description
Example15: Sentence: "William Shakespeare's plays, such as 'Hamlet' and 'Macbeth,' are considered masterpieces of English literature. They explore complex themes like ambition, revenge, and madness." Label: Description

**SUBJ**

Please perform Subjectivity Classification task. Given the sentence, assign a label from ['subjective', 'objective']. Return label only without any other text.
Example1: Sentence: "The soundtrack was moving." Label: subjective
Example2: Sentence: "The soundtrack added an emotional depth." Label: subjective
Example3: Sentence: "The pacing started slow. It built tension. The climax felt rushed." Label: subjective
Example4: Sentence: "The visual effects were impressive. However, a few scenes felt overdone. Enhanced the world-building but occasionally distracted from the story." Label: subjective
Example5: Sentence: "The pacing was uneven. The first half dragged while the second felt rushed." Label: subjective
Example6: Sentence: "The visual effects were impressive, though a few scenes felt overdone. They enhanced the world-building but occasionally distracted from the story." Label: subjective
Example7: Sentence: "The pacing started slow but built tension." Sentence: "Climax felt rushed." Sentence: "Overall, uneven." Label: subjective
Example8: Sentence: "The lead actor brought depth to the role." Label: subjective
Example9: Sentence: "The pacing started slow. It built tension. The climax felt rushed." Label: subjective
Example10: Sentence: "The lead actress gave a nuanced performance." Label: subjective
Example11: Sentence: "The editing was choppy, disrupting the flow. It felt rushed at times." Label: subjective
Example12: Sentence: "The screenplay was tightly constructed. The dialogue flowed naturally. Characters spoke authentically, enhancing the plot." Label: subjective
Example13: Sentence: "The screenplay was tight. The dialogue flowed naturally. Characters spoke authentically, enhancing the plot. Subtle hints of conflict kept the audience engaged." Label: subjective
Example14: Sentence: "The soundtrack added an emotional depth. However, the choice of music was sometimes jarring." Label: subjective

# H  SIMPLIFICATION PROMPT

In this section, we present the best-performing prompts for PIAST on the simplification tasks. The base prompts are adapted from Guo et al. (2023), while the examples are generated using our method.

**SIMPLIFICATION**

Simplify the text.
Example1: Complex: "The Supreme Court decision declared the law unconstitutional, invalidating it." Simple: "The Supreme Court declared the law unconstitutional."
Example2: Complex: "Mount Everest is the highest mountain in the world located in the Himalayas." Simple: "Mount Everest is the highest mountain in the Himalayas."
Example3: Complex: "The pizza place offers a variety of toppings including pepperoni and mushrooms." Simple: "The pizza place offers pepperoni and mushrooms."
Example4: Complex: "The Eiffel Tower is a famous landmark in Paris, France." Simple: "The Eiffel Tower is in Paris, France."
Example5: Complex: "The university offers a range of degree programs in various fields of study." Simple: "The university offers degree programs."
Example6: Complex: "The student passed the exam with excellent grades." Simple: "The student passed with excellent grades."
Example7: Complex: "The basketball game was won by the team with the highest score at the end of the game." Simple: "The team with the highest score won."
Example8: Complex: "The Renaissance was a period of great cultural change and achievement that started in Italy in the 14th century." Simple: "The Renaissance started in Italy in the 14th century."
Example9: Complex: "The Industrial Revolution began in the late 18th century and changed manufacturing methods." Simple: "The Industrial Revolution changed manufacturing methods in the late 18th century."
Example10: Complex: "The Mona Lisa is a famous painting by Leonardo da Vinci." Simple: "The Mona Lisa is a famous painting."
Example11: Complex: "The internet is a global network that connects computers and allows for communication." Simple: "The internet connects computers for communication."
Example12: Complex: "The Supreme Court ruled that the law was unconstitutional." Simple: "The Supreme Court said the law was unconstitutional."
Example13: Complex: "The Renaissance art focused on humanism and realism, emphasizing individual expression and naturalism." Simple: "Renaissance art emphasized individual expression."
Example14: Complex: "The train arrived late due to a track problem." Simple: "The train was late due to a track problem."
Example15: Complex: "The internet protocol is a set of rules that allows computers to communicate over the internet." Simple: "The internet protocol allows computers to communicate."

# I    SUMMARIZATION PROMPTS

In this section, we present the most effective prompt for PIAST on summarization tasks. The base prompts are adapted from Guo et al. (2023), and the examples are generated by our method.

**SUMMARIZATION**

How would you rephrase that in a few words?
Example1: Text: ""IBM is an American multinational technology company headquartered in Armonk, New York."" Summary: ""IBM is headquartered in New York.""
Example2: Text: ""Apple Inc. is an American multinational technology company headquartered in Cupertino, California."" Summary: ""Apple Inc. is headquartered in California.""
Example3: Text: ""Tesla is an American electric vehicle and clean energy company."" Summary: ""Tesla is an electric vehicle company.""
Example4: Text: ""The NBA All-Star Game is an annual basketball game featuring the top players from the National Basketball Association."" Summary: ""The NBA All-Star Game features top NBA players.""
Example5: Text: ""Global warming is caused by an increase in greenhouse gases, leading to rising temperatures and climate changes."" Summary: ""Global warming is caused by rising greenhouse gas levels.""
Example6: Text: ""Tesla is an American electric vehicle and clean energy company founded by Elon Musk, known for its innovative electric cars and energy storage solutions."" Summary: ""Tesla is an electric vehicle company.""
Example7: Text: ""The Supreme Court justices are appointed by the President and confirmed by the Senate, serving life terms."" Summary: ""Supreme Court justices are appointed by the President and confirmed by the Senate.""
Example8: Text: ""The Supreme Court of the United States is the highest court in the country, responsible for interpreting the Constitution and ensuring federal laws are followed."" Summary: ""The Supreme Court of the United States interprets the Constitution.""

Example9: Text: ""The Mona Lisa, a painting by Leonardo da Vinci, is one of the most famous and most visited paintings in the world, currently housed in the Louvre Museum in Paris."" Summary: ""The Mona Lisa is a famous painting by Leonardo da Vinci.""

Example10: Text: ""The Louvre Abu Dhabi, opened in 2017, is a museum in Abu Dhabi that focuses on art and culture from around the world."" Summary: ""The Louvre Abu Dhabi opened in 2017 and focuses on global art and culture.""

Example11: Text: ""The Louvre Museum in Paris is one of the largest and most visited art museums in the world, with a vast collection of art and artifacts."" Summary: ""The Louvre Museum in Paris has a large art collection.""

Example12: Text: ""The American Revolution was a violent conflict between Great Britain and thirteen of its North American colonies from 1775 to 1783."" Summary: ""The American Revolution lasted from 1775 to 1783.""

Example13: Text: ""The Renaissance was a period of great cultural and intellectual growth in Europe, spanning the 14th to the 17th century."" Summary: ""The Renaissance was a period of cultural and intellectual growth in Europe.""

Example14: Text: ""The Renaissance was a period of great cultural and intellectual growth in Europe, spanning the 14th to the 17th century, marked by a revival of classical learning."" Summary: ""The Renaissance was a period of cultural and intellectual growth in Europe.""

Example15: Text: ""The Eiffel Tower is a wrought-iron lattice tower on the Champ de Mars in Paris, France."" Summary: ""The Eiffel Tower is in Paris, France.""

Example16: Text: ""The giant panda is a bear species endemic to central China, recognized by its distinctive black and white fur and diet primarily consisting of bamboo."" Summary: ""The giant panda is a bear species endemic to China.

## J    USE OF LLMS

Large Language Models (LLMs) were employed in this paper for refining and polishing the writing.

