# OpenReview forum: "PIAST: Rapid Prompting with In-context Augmentation for Scarce Taining data"
_ICLR.cc/2026/Conference — ICLR 2026 Conference Withdrawn Submission_

### Official Review · Reviewer_rmAh · 2025-10-15

**Soundness:** 2
**Presentation:** 2
**Contribution:** 2
**Rating:** 2
**Confidence:** 5

**Summary:**

This paper proposes a new framework for optimizing few-shot prompts for LLMs. Specifically, they propose to use MC Shapley estimation to evaluate the importance of each demonstration. And they also use a replay buffer for their evaluation. They call their framework PIAST. The results on several NLP benchmarks including Classification, Simplification and Summarization show that their framework can be comparable to existing state-of-the-art baselines including APE, APO, EvoPrompt and PRL.

**Strengths:**

Using Shapley values for demonstration selection is a novel idea.

Prompt optimization and in-context learning merit further study.

**Weaknesses:**

1. The writing shall be significantly improved. First of all, I am not quite sure what you mean by "compute time budget" in the abstract. This word can look ambiguous. And, for the Table 1, I am not convinced enough about your speed column here. This is quite subjective, without any concrete evidence, at least to me. Readers wouldn't be able to understand what it implies.

2. I suspect that this paper misses some baselines, which are crucial. For example, as your main point is to optimize few-shot prompts, some very recent **published work** regarding ICL selection shall be compared. Otherwise, it is hard for us to position your work.

To be honest, I indeed can see a lot, such as PromptBreeder in ICML 2024 and PromptQuine in ICML 2025, and many others which release their codebases.

And maybe it is also helpful to compare against some (zero-shot --- no demos) prompt optimization frameworks, e.g., RLPrompt which is quite famous in this domain. This helps position your work.

And I am not convinced enough regarding how you implement the baselines. Some baselines can also be applied in many prompt optimization settings (e.g., w/ or w/o few-shot demos). But obviously, it is underspecified in your paper in elucidating the details. I am not convinced enough regarding the numbers. Slight variations could lead to huge differences in search problems.

You should address this and discuss this clearly in the paper.

3. And to be honest, the improvement over other baselines is not quite significant. Others may criticize that this paper may lack some interesting insights. If you want to emphasize the strengths of your search speed, you shall provide more visualizations and discussions in the paper, then readers can feel easier to understand. The current version doesn’t work well at least in this lens. I see many distracting things instead.

**Questions:**

See my weaknesses.

---

### Official Review · Reviewer_jYvv · 2025-10-31

**Soundness:** 3
**Presentation:** 2
**Contribution:** 2
**Rating:** 4
**Confidence:** 4

**Summary:**

his paper proposes PIAST, a fast, automatic prompt construction algorithm designed to overcome the high time and data requirements of existing methods. PIAST augments a concise, manually-written instruction with a small set of automatically generated in-context examples. The optimization is handled through an iterative improve-loop (replace/drop/keep) that uses Monte Carlo Shapley estimation to efficiently determine and optimize the most valuable examples. By incorporating aggressive subsampling and LLM serving optimizations (e.g., KV-Cache reuse), the method achieves high efficiency. PIAST outperforms prior automatic methods on text simplification and GSM8K, and sets a new SOTA for classification, proving that carefully constructed in-context examples are the key lever for fast and data-efficient prompt engineering.

**Strengths:**

1.  PIAST is the first method to claim both fast execution and the ability to synthesize new, high-quality in-context examples, solving the limitations of prior slow (PRL) or reuse-only (APO) techniques.

2. The use of Monte Carlo Shapley values is a sophisticated and robust mechanism for example selection, effectively capturing the marginal utility and interdependence of examples better than simpler methods like LOO.

3. The method achieves SOTA/near-SOTA results on multiple tasks with significantly less time and data than competitors, making it highly practical for resource-limited settings.

**Weaknesses:**

1. The novelty of PIAST is limited. There are many existing works focusing on the synthesis, selection, and optimization of in-context examples. The authors should clearly demonstrate the differences between PIAST and these methods, such as PromptWizard and SEE (Strategic Exploration and Exploitation for Cohesive In-Context Prompt Optimization).

2. The comparative methods used in the experiments, such as APE, APO, GA, and GE, primarily focus on optimizing the instruction and were not specifically designed to optimize examples. The authors' experiments should include a comparison with few-shot prompt optimization works like PromptWizard and SEE.

3. A comparison between PIAST and other methods regarding token consumption and the number of API calls is crucial.

**Questions:**

See Weakness

---

### Official Review · Reviewer_zAAQ · 2025-10-31

**Soundness:** 4
**Presentation:** 3
**Contribution:** 3
**Rating:** 6
**Confidence:** 3

**Summary:**

This paper proposes PIAST, an automatic prompt construction algorithm that augments a short human instruction with a small set of synthetically generated few-shot examples, then iteratively improves that example set using a Monte Carlo Shapley estimator to replace / drop / keep examples. The method is designed for data and time-constrained settings: it aggressively subsamples evaluation data, keeps a replay buffer to avoid overfitting, and uses KV-cache reuse, paged attention, and continuous batching to stay fast. On classification, summarization, simplification, and GSM8K, PIAST matches or beats prior automatic prompting methods (APE, APO, EvoPrompt, PRL), and with a slightly larger budget (PIAST-E) it reaches new SoTA among automatic prompting methods on several tasks. The paper’s central claim is that iteratively refined in-context examples, not exhaustive instruction search, can achieve faster, more data-efficient prompt engineering.

**Strengths:**

-- Using Monte-Carlo approximation of Shapley values is an intuitive way to capture redundancy/complementarity compared to the simple Leave-on-out metric.

-- Comprehensive comparison to different baselines and different variations of PIAST (different runtime budget, access to training data).

-- Evaluation is performed across a wide range of traditional NLP tasks (various classification, sentence simplification, summarization) and grade-school math. The proposed method achieves competitive performance across all benchmarks.

-- Comprehensive ablation studies on different components/algorithmic choice of PIAST.

**Weaknesses:**

**-- Method section structure:** The Method section is poorly structured and is confusing at the first glance. The authors should use a hierarchical structure (e.g., subsections) rather than parallel paragraphs for all the topics. For example, Example Selection via Shapley Values (line 175-193), Monte-Carlo Approximation (line 200-207), and Replace/Drop/Keep decision (line 209-225) are the components of Example Improver. This hierarchy is not reflected from the structure and requires extra effort from the authors to confirm it.

**-- LLM homogeneity:** almost everything is done with Qwen2.5-7B for all 3 roles; cross-model transfer is only lightly tested and shows some overfitting for PIAST-E. This weakens the “general” claim.

**-- Limited ablation on example generator quality:** PIAST’s success may depend heavily on how good the initial synthetic examples are. The paper could be more complete with an analysis when the base LLM is weaker or the task is very structured.

**Questions:**

-- How does PIAST behave when the base LLM is not strong enough to synthesize diverse, high-quality examples? Does performance collapse or just plateau? Did you try PIAST on smaller models (e.g., Qwen2.5-0.5B)?

-- The replay buffer is presented as a regularizer. But how large does it need to be before gains saturate, and does it ever hurt by anchoring to early, suboptimal data subsets?

-- Could PIAST be combined with instruction rewriting (APE-style) in the same loop, or would that break the KV-cache and runtime assumptions?

-- For GSM8K, can you show accuracy vs #examples in the final prompt, to confirm the claim that example quality (not just count) is the real driver?

---

### Official Review · Reviewer_6Whj · 2025-10-31

**Soundness:** 2
**Presentation:** 2
**Contribution:** 2
**Rating:** 2
**Confidence:** 4

**Summary:**

This paper proposes PIAST, an automatic prompt construction algorithm that augments a concise instruction with a small, iteratively refined set of few-shot examples. The core idea is to treat in-context examples as the primary optimization lever and to update them via a replace/drop/keep loop guided by a Monte-Carlo approximation to Shapley values computed over ordered subsets of examples.

**Strengths:**

1. The replay buffer and batching/KV-reuse make the method usable under tight budgets.
2. On GSM8K and simplification, PIAST outperforms instruction-rephrasing baselines and is competitive with PRL at a fraction of the runtime.

**Weaknesses:**

1. The introduction underrepresents prior work on few-shot example optimization and automatic example selection [1]. The claim that optimizing in-context examples is underexplored is overstated.
2. The claim in lines 54–55 that PIAST uniquely generates examples “not found in the training set” is inaccurate, as prior baselines like PRL already synthesize novel example pairs. This makes the contribution appear less novel than implied.
3. The performance of PIAST heavily relies on the capacity and alignment of the Example Proposer, which generates the initial candidate examples. When the proposer is weak or biased, downstream refinement has limited effect. The paper does not analyze sensitivity to proposer quality or cross-model substitution.
5. The Prompt Evaluator is optimized on small validation subsets, and the replace/drop/keep policy depends on stochastic evaluations plus a replay buffer. This setup risks overfitting to transient evaluator preferences. Although the replay buffer mitigates drift, the paper provides no variance analysis or confidence intervals for selection stability, leaving the robustness of the search process unclear.
6. Despite the broad task sets, evaluations focus on relatively easy benchmarks and short inputs. It remains unclear how PIAST works for long-context and challenging tasks.
7. The same backbone model family (Qwen2.5-7B-Instruct) is used for proposing, evaluating, and final inference. This tight coupling raises concerns about model-specific overfitting. The paper includes minimal analysis of whether the resulting prompts transfer effectively to other model families.
8. The reported improvements are modest given the multi-agent design. The extended variant PIAST(E) offers additional gains while incurring substantial compute cost and potential overfitting to the source evaluator. This trade-off undermines the claimed fast and efficient appeal.


[1] Automatic Chain of Thought Prompting in Large Language Models

**Questions:**

For larger LLMs (e.g., 32b), which already perform robustly in few-shot settings, it is uncertain whether PIAST’s multi-agent refinement offers meaningful gains or simply introduces redundant complexity. No evidence or scaling discussion is provided.

---

### Official Review · Reviewer_rRTV · 2025-10-31

**Soundness:** 3
**Presentation:** 4
**Contribution:** 2
**Rating:** 4
**Confidence:** 4

**Summary:**

This paper introduces PIAST, a fast automatic prompt construction algorithm that augments human instructions by generating a small set of few-shot examples. The method iteratively refines this example set using a replace/drop/keep loop , where the utility of each example is determined by a Monte Carlo Shapley estimation. Experiments show PIAST achieves state-of-the-art or highly competitive results on text classification, simplification, and GSM8K , while being substantially faster and more data-efficient than existing automatic prompting methods.

**Strengths:**

1. Synthetic in-context example generation. Existing approach mainly selects demonstration from training or modify instruction while this work generates synthetic examples that could target the failure cases.
2. I find the experiments and ablation study is thorough and comprehensive. It is informative for me to understand more than just the results.

**Weaknesses:**

1. The speed up appears to reply on the engineering tricks rather than the method itself, which make it impossible for the speed advantage transferring to other scenarios where engineering tricks does not apply. The setup is even stronger than white box model. It could be good to understand the speed comparison under the same setup without modifying model inference infra.

2. Although the paper bears novelty and inspiring results from few-shot example manipulation and synthetic sample generation, the novelty on prompt optimization is limited.

3. In-context demonstration empirically has positive relation with task performance. The authors presents analysis on the effect of number of demonstration samples, and includes that demonstration count affects performance remarkably. The comparison setup on few-shot numbers is not very clear in the paper. It is unclear to me whether this effect is controlled for different baselines in main experiments.

**Questions:**

The questions are mainly based on the weakness session:

1. How many demonstration examples do you use in main results? How about other baselines? Do they all require in-context examples?
2. Could you present speed comparison from purely token cost rather than system latency since the latter is highly affected by infra and inference setup.
3. The baseline you consider includes the ones that directly optimize the instruction, which should be a more direct way to handle failure cases, why do you think in-context examples (indirect way) handle them better and give better performance?

---

### Note · Authors · 2025-12-22

**Comment:**

We thank the reviewers for their time and thoughtful feedback.

**Withdrawal Confirmation:**

I have read and agree with the venue's withdrawal policy on behalf of myself and my co-authors.